# Double-Center-Based Iris Localization and Segmentation in Cooperative Environment with Visible Illumination

**DOI:** 10.3390/s23042238

**Published:** 2023-02-16

**Authors:** Jiangang Li, Xin Feng

**Affiliations:** College of Computer Science and Engineering, Chongqing University of Technology, Chongqing 400054, China

**Keywords:** iris localization, iris segmentation, center-based object detection

## Abstract

Iris recognition has been considered as one of the most accurate and reliable biometric technologies, and it is widely used in security applications. Iris segmentation and iris localization, as important preprocessing tasks for iris biometrics, jointly determine the valid iris part of the input eye image; however, iris images that have been captured in user non-cooperative and visible illumination environments often suffer from adverse noise (e.g., light reflection, blurring, and glasses occlusion), which challenges many existing segmentation-based parameter-fitting localization methods. To address this problem, we propose a novel double-center-based end-to-end iris localization and segmentation network. Different from many previous iris localization methods, which use massive post-process methods (e.g., integro-differential operator-based or circular Hough transforms-based) on iris or contour mask to fit the inner and outer circles, our method directly predicts the inner and outer circles of the iris on the feature map. In our method, an anchor-free center-based double-circle iris-localization network and an iris mask segmentation module are designed to directly detect the circle boundary of the pupil and iris, and segment the iris region in an end-to-end framework. To facilitate efficient training, we propose a concentric sampling strategy according to the center distribution of the inner and outer iris circles. Extensive experiments on the four challenging iris data sets show that our method achieves excellent iris-localization performance; in particular, it achieves 84.02% box IoU and 89.15% mask IoU on NICE-II. On the three sub-datasets of MICHE, our method achieves 74.06% average box IoU, surpassing the existing methods by 4.64%.

## 1. Introduction

Iris recognition is one of the most reliable biometric technologies, and is widely applied in intelligence unlocking, border control, and forensics, among others [1,2,3,4]. A complete iris-recognition system usually consists of image acquisition, iris segmentation and localization, normalization, feature extraction, and matching. Figure 1 illustrates the key steps for iris-image preprocessing, which contains location, segmentation, and normalization. Iris localization aims to detect the inner and outer boundaries of an iris region; iris segmentation generates an iris mask to distinguish iris and non-iris pixels. The normalization process allows the alignment of any two iris images to be compared. After image preprocessing, we can achieve the normalized iris texture image and iris/noise mask image; therefore, an important part of the iris recognition pre-processing, iris segmentation and localization jointly define the region used for feature extraction and matching, directly affecting the overall iris-recognition performance.

Most previous studies have focused on iris images in cooperative environments (e.g., near-infrared illumination, high user collaboration, close acquisition distance, and stop-gaze verification), and two widely used localization methods are Daugman’s integro-differential operator [5] and Wilde’s circular Hough transforms [6]. Recently, it has been extended to non-cooperative environments (e.g., long distances, limited user cooperation, visible lighting, and mobile devices) because it requires minimally restrictive user cooperation and imaging conditions; however, iris images captured in non-cooperative environments often have various noise, such as gaze bias, iris rotation, aniridia reflection, specular reflection, motion/defocus blur, and eyelid/lash/hair/glasses occlusion (see Figure 2), thereby making iris localization challenging.

Recent deep learning-based methods use convolution neural networks (CNNs) to separate the iris in the image and use circular Hough transform [6] thereafter to fit the inner- and outer-circle parameters on the iris mask. These methods have achieved good results in iris mask segmentation, but localization is extremely rough and accuracy remains below the standard required for iris recognition. Other CNN-based methods, such as IrisParseNet [7] and NIR-Zhang [8], use the semantic segmentation method to predict inner-/outer-circle boundary masks, and utilize some post-processing thereafter to obtain parameterized inner and outer circles. Although localization is improved, it requires massive refined post-processing, and the accuracy of the inner- and outer-circle localizations is easily influenced by a fuzzy contour boundary.

We use the preceding issues as bases in considering the following three aspects: (1) how toachieve a more accurate and robust localization method on iris images captured in a non-cooperative environment with visible illumination? (2) performing post-processing (e.g., integro-differential operator-based or circular Hough transforms-based method) on pupil or iris mask to fit the iris’ inner and outer circle parameters (x,y,radius) is time-consuming and not robust, can the model directly output the localization results of the inner and outer circles without any post-processing? (3) On the basis of (2), can we obtain a simple but effective end-to-end model that integrates localization and segmentation in one network? For the above three aspects, and considering the natural biological characteristics of the iris region, we propose a double-center-based method to localize and segment the iris region. Our main contributions are summarized as follows:(1)We propose to locate the iris’s inner and outer circles as center points and regression radius on feature maps directly, thereby solving the problems of prediction inaccuracies and lacking robustness of iris localization on the visible iris image. Compared with existing methods that use massive post-processing on the predicted mask to obtain the circle parameters, our approach is post-processing-free and the localization module output is the final inner-/outer-circle localization result;(2)By explicitly analyzing the distribution of the center points of the pupil and iris, we propose a novel auxiliary sample strategy to accelerate model training on non-standard iris images. These images have irrelevant face regions, such as chin, nose, and environments;(3)We design an end-to-end dedicated iris localization and segmentation framework, which is simple, effective, and achieves excellent performance in multiple benchmarks. The proposed method provides a good foundation for iris localization and segmentation in a non-cooperative environment.

The remainder of this paper is organized as follows: Section 2 introduces the related work on iris segmentation and localization. Section 3 presents details pertaining to the proposed iris localization and segmentation methods. Section 4 presents ablation, and compares and analyzes the experimental results. Section 5 summarizes the research and concludes the study.

## 2. Related Research

Most traditional iris-segmentation methods mainly use a circle or ellipse to locate the inner and outer boundaries of the iris, and then use the difference of the gray histogram to exclude any superimposed occlusions of eyelashes, shadows, glasses, or reflections, and infer the iris region [3]. 

For iris localization, two widely used baseline methods are Daugman’s integro-differential operator [5] and Wildes’s Hough transforms [6]. The integro-differential operator searched for the largest difference of intensity over the circle parameter space, and it has achieved great precision; however, it may take a relatively longer time [9]. The circular Hough transforms found optimal circle parameters by a voting procedure in a binary edge image, and is often applied to detect boundaries of the circle or ellipse, but this transform is relatively non-sensitive to the broken contours of 2D objects in the binary image [10]. 

Based on the above two basic methods, many later proposed approaches made further improvements in accuracy and efficiency. For example, ref. [11] applied *L*_1_ norm to suppress noisy texture before performing iris localization, ref. [12] applied region clustering before localization for narrowing the parameter search range, ref. [13] proposed integro-differential constellation to reduce the computation time, and ref. [14], applied the Viterbi algorithm on gradient maps of iris images to find coarse low-resolution contours. Those traditional methods rely considerably on prior assumptions and image low-level features, so they have low accuracy and poor robustness in dealing with noise.

Recently, deep learning-based the semantic segmentation method has achieved higher accuracy in iris segmentation. Different from the traditional pixel-based iris-segmentation method, the deep learning-based iris-segmentation methods utilize high-semantic features and estimate iris masks end to end. The first CNNs-based iris-segmentation method is HCNNs [15]. Other iris-segmentation methods based on CNNs include those in [16,17,18,19,20,21]. These methods use convolution to extract semantic features and segment the foreground of the iris, thereby achieving good iris mask segmentation. 

Although CNNs-based iris-segmentation method has achieved good results, most of those methods only predict iris masks, and more important iris localization is not achieved. Ref. [15] first applied the circular Hough transforms method from original iris image to iris masks to generate candidate circular iris boundaries, and then use two quality measures to select the best inner and outer iris boundaries.

Current CNN-based methods first segment the image to achieve the iris mask and utilize a massive post-process, such as Daugman’s integro-differential operator and Hough transform, to fit the inner and outer circles on the iris mask. As shown in Figure 3a, similar methods are also presented in [22,23]. These methods work well on iris images taken in a cooperative environment (e.g., near-infrared illumination, high user collaboration, close acquisition distance, and stop-gaze verification), but in iris images taken in non-cooperative and visible light environments, the performance was mediocre. IrisParsetNet [7] first uses the segmentation method to predict the inner and outer boundaries of the iris. While the model predicts the iris mask, it also simultaneously predicts the pupil and iris outer boundary masks, as shown in Figure 3b. NIR-Zhang [8] uses two completely independent models responsible for iris segmentation and iris localization, as shown in Figure 3c. The localization model predicts pupil and iris masks and uses post-processing thereafter to fit the contour and circle parameters.; however, the massive post-process to fit a circle on the boundary mask is time-consuming and not robust on the iris images captured in non-cooperative environments with visible illumination. The reason is that the localization results are easily affected by irregular contour edges.

To obtain an efficient post-processing-free iris-localization solution and achieve a more accurate and robust iris localization on iris images captured in a non-cooperative environment, we propose a center-based iris localization and segmentation method and make localization rely on a feature map other than the iris mask (see Figure 3d). Inspired by the use of heat-map for face key point detection [24], we regard the inner/outer circles as two objects to locate. We first locate the double center and regress the radius thereafter based on the center. A segmentation branch is embedded after the location module, which is responsible for segmenting the iris and background pixels to obtain the iris mask. Our method is fast, accurate, and does not need any post-processing. The model’s output is inner/outer circle $\left(x_{k},y_{k},\;r_{k}\right),k\in \left[inner,\;outer\right]$ and iris mask directly.

## 3. Methods

The architecture of the proposed ICSNet, as shown in Figure 4, mainly includes the localization and segmentation modules. The localization module consists of two subnets, namely, classification and radius regression heads, which are responsible for predicting the inner- and outer-circular bounding boxes. The segmentation module consists of RoI crop and the mask head, which are accountable for segmenting the iris mask over the detected region. As an auxiliary structure, the proposed concentric sample strategy only exists during the training stage and is cost-free in inference time.

First, this section introduces the localization module, which is our model’s essential iris-localization component. Second, we present the mask-segmentation branch, which follows the localization module. Lastly, we introduce the sample strategy.

### 3.1. Localization Module

#### 3.1.1. Double-Center Localization

Current iris localization uses iris or contour mask to fit the inner/outer circles, and these fitting methods can be divided into integro-differential operator-based and circular Hough transforms-based methods; however, owing to the noise in a non-cooperative environment with visible light, these methods are susceptible to irregular mask or contour edges, resulting in poor robustness and inaccuracy. To address these issues, we propose a double-center-based iris-localization method, and the model relies on a feature map to directly predict the iris’s inner and outer circles.

Given an input image $I\in R^{W*H*3}$, we adopt the Gaussian kernel as in CornerNet [25] to produce a ground truth heat map $Y_{x,y,c}\in {\left[0,1\right]}^{W*H*C}$, where (x,y) is the location and C is the number of center point categories. In our experiment, we set C=2 (inner or outer circle). The heat map and input have the same size because our backbone is an encoder–decoder fully convolution network. A prediction ${\hat{Y}}_{x,y,c}=1$ corresponds to a detected center point, while ${\hat{Y}}_{x,y,c}=0$ is background.

For each ground truth center point $\left(p_{x},\;p_{y}\right)$, we compute an equivalent $\bar{p}=\lfloor p\rfloor$; thereafter, we splat center points $\left({\bar{p}}_{x},\;{\bar{p}}_{y}\right)$ onto a heat map using a Gaussian kernel, which is the same as [26]. The peak of the Gaussian distribution is treated as a positive sample, while another pixel is treated as a negative sample. We adopt modified focal loss [27] to train.

The localization module follows behind the backbone. As shown in Figure 4, the module consists of three convolution layers and sigmoid function. The localization module predicts two heat maps, with all value range in [0,1], and the location of the maximum of one value in each heat map is the predicted center point $\left(x_{k},y_{k}\right),k\in \left[inner,outer\right]$.

#### 3.1.2. Radius Regression

Owing to the unique physiological characteristics of the iris region, the centers of the inner and outer circles appear in a pair; therefore, we set up two branches to be responsible for the inner- and outer-circle radius regressions.

Let $p_{k}=\left(x_{k},\;y_{k}\right)$ be the circle box center points, where k represents the center point category (inner or outer circle). We utilize heat map $\hat{Y}$ to predict center point $p_{k}$, and regress circle radius $r_{k}$ thereafter for each center point $p_{k}$. We adopt an *L*_1_ loss to train, which is defined as follows:(1)$$\begin{array}{c} L_{size}=\frac{1}{2}\overset{2}{\underset{k=1}{\sum }}|{\hat{r}}_{p_{k}}-r_{k}|.\; \end{array}$$

Thereafter, we directly use raw pixel coordinates for regression. To make the regression more accurate, we regard the samples in the Gaussian distribution region as positive samples, which are all responsible for predicting the object radius. Samples outside the Gaussian region are considered negative samples, and these samples do not need to predict the radius.

Additionally, we predict an offset for the center point to recover the discretization error caused by the integer operation and output stride. The offset is training with an *L*_1_ loss. We take the pixel in the center of the object to predict the offset value directly, which is defined as follows:(2)$$\begin{array}{c} L_{reg}=\lambda_{size}L_{size}+\lambda_{off}L_{off}.\; \end{array}$$

In our experiments, we set $\lambda_{size}=1$ and $\lambda_{off}=0.1$, following [28].

### 3.2. Segmentation Module

After the localization module process, we obtain the inner and outer center $\left({\hat{x}}_{c},\;{\hat{y}}_{c}\right)$ and its radius ${\hat{r}}_{c}$. According to the detected center point of the outer circle, we crop the $\hat{r}\times \hat{r}$ RoI region on the feature map as the mask head input. We select only one iris RoI region according to the center point of the outer circle with confidence, thereby avoiding time-consuming post-processing. 

The mask head comprises three convolution layers and sigmoid function. Each convolution layer consists of one 3 × 3 convolution layer, in which the stride is 2 and padding is 1, ReLU activation, and batch normalization (see Figure 4). After the mask branch process, we obtain the mask of the iris in the RoI region. The mask branch adopts the cross-entropy loss function to train.

### 3.3. Total Loss

Total loss *L* is composed of localization loss $L_{cls}$, regression loss $L_{reg}$, and mask loss $L_{mask}$, and weighted by three scalars, defined as follows:(3)$$\begin{array}{c} L=w_{loc}L_{cls}+w_{reg}L_{reg}+w_{mask}L_{mask},\; \end{array}$$
where $L_{cls}$ is based on the modified focal loss [28], $L_{reg}$ is calculated according to Equation (2), and $L_{mask}$ adopts the cross-entropy loss. In our experiment, we set $w_{loc}$, $w_{reg}$, and $w_{mask}$ as 1, 1, and 1, respectively.

### 3.4. Backbone

The backbone network of the model is an encoder–decoder full convolution network, which can accept any input size; moreover, the backbone is based on deep layer aggregation (DLA) [29]. In the encoder, we use a basic structure block in DLA. In the decoder, we use the up-sampling of bilinear interpolation and a single convolution layer, consisting of one 3 × 3 convolution layer with stride 2, ReLU activation, and batch normalization. The depth of the decoder and encoder is 5. We use a shortcut to connect the encoder block and decoder layer. The output feature map has 32 channels.

### 3.5. Concentric Sampling Strategy

Given the natural biological characteristics of the iris region, the centers of the inner and outer circles would overlap. Hindered by the camera device and non-cooperative environment, two centers in the captured image occasionally have a slight shift, as shown in Figure 5a,b, and annotated circles; moreover, the Gaussian heat map has overlapping regionsas shown in Figure 5c; therefore, we define a concentric Gaussian region (CGR) as the inner and outer circles sharing the same Gaussian value, which is represented as follows:(4)$$\begin{array}{c} Y_{i,j}=\{\begin{array}{c} 1,\;\;\;if\;Y_{i,j,inner}=1\;and\;Y_{i,j,outer}=1\\ 0,\;\;\;if\;Y_{i,j,inner}=0\;or\;Y_{i,j,outer}=0\\ min\left(0,5,\frac{1}{2}\sum \alpha_{c}Y_{i,j,c}\right),\;else \end{array}, \end{array}$$
where $Y_{i,j,c}$ presents the original Gaussian distribution, in which the calculation method is the same as [30], c presents the category (inner or outer), and α is the hyper-parameter, presenting the weight of the inner and outer Gaussian distribution in the overlapping region; thereafter, the value is clipped to the range [0,0.5]. Locations of the inner/outer center points and their four directions are high activation points; hence, we set these values in the heat map as 1.0 and 0.8, respectively.

For the Gaussian distribution, we set the pixel at the circular box center region as the positive sample, while another pixel is the negative sample. We limit the inner Gaussian distribution and outer Gaussian distribution to the inner-circle region, avoiding the imbalance of loss contribution caused by the Gaussian distribution area and making the model focus immediately on important regions. The concentric sampling strategy is completely cost-free in inference time, as the auxiliary structures only exist during the training stage.

### 3.6. Double Center to Circular Boxes

During the inference, the model outputs 2 heat maps ${\hat{Y}}^{H\times W\times 1}$, where the values indicate the confidence scores of the inner and outer centers. The confidence score is sorted, and the coordinate position corresponding to the maximum value is the predicted target center point $\left(x_{k},\;y_{k}\right),\;k\in \left[inner,\;outer\right]$. On the basis of the center point, the corresponding radius and offset can be directly derived.

For the center point position (x,y) and predicted radius $\hat{r}$ and offset $\left(\delta_{x},\delta_{y}\right)$, the corresponding circular bounding box coordinates are calculated as follows:(5)$$\begin{array}{c} \begin{array}{c} \hat{x}=x*s+\delta_{x}\;\\ \hat{y}=y*s+\delta_{y} \end{array},\; \end{array}$$
where s is the scale ratio of the feature map and input size. Owing to the particularity of the backbone network, in the experiment, we set s=1; therefore, the predicted radius is the size of the original pixel of the image, and $\left(\hat{x},\hat{y},\hat{r}\right)$ is the final predicted result.

## 4. Experiments

### 4.1. Experimental Environment

We use Adam optimizer, and the model weight is initialized randomly. The initial learning rate is 0.001, weight decay is set as 0.0004, and mini-batch is 6. During training, the learning rate is automatically adjusted, and the total epoch is 120. 

In the inference phase, we resize the input image to a specific resolution and forward it thereafter to directly obtain the predicted circle bounding boxes (inner and outer) and the iris mask.

### 4.2. Data Set and Evaluation Protocols

#### 4.2.1. Data Set

NICE-II [31] comprises two non-overlapping subsets: (i) a training set with 1000 images from 171 subjects and (ii) a testing set with 1000 images from 150 subjects. We use all training sets to train and all testing sets to test. 

MICHE [32] is composed of three sub-databases: GS4, IP5, and GT2. Images are captured in an uncontrolled environment using three mobile devices. MICHE-GS4 has 1297 images (663 and 634 indoor and outdoor images, respectively), IP5 has 1262 images (631 and 631 indoor and outdoor images, respectively), and MICHE-GT2 has 632 images (316 each for indoor and outdoor images). In the experiment, we use all indoor images for training and all outdoor images for testing for each sub-data set. 

For the above four datasets, NICEII, MICHE-GT2, MICHE-GS4, and MICHE-IP5, the picture resolution (width × height) is 300 × 400, 400 × 300, 270 × 444, and 270 × 444, respectively. Since our network is a full convolutional encoder-decoder network, the image resolution that can be entered needs to be a multiple of 32; therefore, the input size of each dataset is 320 × 448, 448 × 320, 320 × 448, and 320 × 448, respectively. In addition to expanding the width and height by central cropping, there is no other data processing.

The images of these databases are captured from user non-cooperative and visible light environments, and taken by mobile devices rather than imaging sensors (such as infrared iris cameras); therefore, these images usually contain light reflection, gaze deviation, defocusing, mirror reflection, and other noises. Some examples of eye images and ground truths are shown in Figure 6. We aim to achieve more accurate localization of iris inner and outer boundaries, as well as eliminate these noises to obtain valid iris pixels.

#### 4.2.2. Evaluation Protocols

We adopt several evaluation protocols for the inner-/outer-circle localization and iris segmentation to evaluate the proposed method. 

(1).Localization. We compute the inner-/outer-circle box IoU ${\mathrm{mIoU}}^{\mathrm{box}}$, which ranges from [0,1]. The closer the value to 1, the better the localization. We also compute the Hausdorff distance, similar to [7]. We add points coordinate normalization, and the range of normalization Hausdorff distances is between [0,1]. The smaller the value, the higher the shape similarity;(2).Segmentation. We use $E1_{\mathrm{mask}}$
and $E1_{\mathrm{norm}}$, which are the mask errors of the iris and normalized iris masks, respectively, which are the same as [7]. The value range is [0,1]. The smaller the value, the better the result. In addition, we use ${\mathrm{mIOU}}_{\mathrm{mask}}$ to evaluate the segmentation performance, and the value range is [0,1]. The larger the value, the better the segmentation result.

### 4.3. Ablation Study

We conduct an ablation study to demonstrate the influence of the concentric sample. Results on the NICE-II [31] and MICHE [32] data sets are shown in Table 1. In the MICHE database, the images have irrelevant face regions, such as chin, nose, and environment, as shown in the middle column of Figure 6. Note that using this auxiliary training strategy brings a relatively significant performance gain of nearly 4.79%; moreover, the results show that making inner- and outer-circle Gaussian distributions share the same Gaussian value on the overlapping region can cause the model to focus on the critical region. Sampling over the central region is suitable for iris localization on non-standard iris images. In the NICE-II database, the images are in the eye region, as shown in the first column of Figure 6. Note that using CGR degrades the model performance. The low similarity in distribution between the NICE-II and MICHE data sets is the main reason. Using CGR on the NICE-II data set introduces many positive samples, which distract the model attention. After canceling CGR, the model only needs to focus on two positive samples (center point of the inner and outer circles), making the model more accurate.

### 4.4. Compared with Other Methods and Discussion

To verify the effectiveness of our model, we compare proposed method ICSNet with other state-of-the-art methods, FCEDN [18], IrisDenseNet [19], FRED-Net [20], IrisParseNet [7], and NIR-Zhang [8]. Table 2 and Table 3 provide summaries of the performance comparison of the proposed approach with baseline method on iris localization and iris segmentation using the proposed evaluation. We also report the parameter amount and FLOPs in order to further evaluate the proposed approach.

For localization, FCEDN [18], IrisDenseNet [19], and FRED-Net [20] use an integro-differential operator or circular Hough transform to fit the inner and outer circles on the predicted iris mask. IrisParseNet [7] predicts the iris mask and two iris edge masks, and then adopts the circle fitting method on the edge mask to obtain the circular inner and outer iris boundaries. NIR-Zhang [8] use an independent model to predict the inner contour and the outer contour, and then use a circular Hough transform to fit the inner circle and the outer circle. As described in Section 3.1, our method relies on a feature map to predict the inner and outer circles directly.

It can be observed that the IrisParseNet [7] and NIR-Zhang [8] consistently outperforms the other three methods in four databases, in particular, three noisy visible MICHE databases, which show that fitting the circle on the contour mask is better than fitting it on the iris mask; moreover, our approach achieved better localization results on all databases than the above five methods. On the NICE-II, ICSNet achieves 84.02% box mIoU, outperforming other methods by nearly 1.02%; on the MICHE, ICSNet outperforms other methods by nearest 9.28% avg box mIoU, which all reflect that predicting the iris circle on the feature map is more accurate and robust.

The parameters and FLOPs for different methods (resolution: 320 × 320 pixels) are shown in Table 4. Note that our model only needs 47.01 GFLOPs. All experiment results show that our model is well balanced in terms of accuracy and speed.

To better compare the iris localization and segmentation results, we select certain challenging samples and present the normalization results. The visualization results of ICSNet and other methods on the different databases are shown in Figure 7. The inner- and outer-circular boundaries are marked by green and red, respectively. If locating the inner and outer circles fails, then the normalized iris result will be empty. Evidently, for the irregular and noisy iris images formed under visible light illumination in a non-cooperative environment, other mask-based fitting methods cannot achieve accurate and robust iris localization; however, our double-center-based localization method can handle this situation. This result indicates that the localization method of dependent feature maps is better than the fitting method of relying on iris/contour masks.

The existing methods use considerable post-processing to fit the inner- and outer-circular parameters on the iris or contour mask. These methods rely substantially on the segmented iris mask instead of the image feature, have weak robustness, and low localization in a non-cooperative environment. Given the effects of illumination, reflection, and noise, the shape of the segmented iris mask is irregular. Our double-center-based localization method relies on feature map rather than the segmented iris or contour mask, and the model directly predicts the inner- and outer-circle positions; therefore, our model has good localization performance and strong robustness on iris images in a non-cooperative environment with visible illumination.

## 5. Conclusions

In this paper, we propose a novel double-center-based efficient iris localization and segmentation network. Unlike prior research utilizing post-process methods (e.g., circle Hough transform or integro-differential) on iris or contour mask to fit the inner and outer circles, ICSNet relies on feature map to directly predict the center and regress the radius; furthermore, we propose a novel auxiliary sample strategy to accelerate model training on non-standard iris images captured in a non-cooperative environment with visible illumination. The proposed approach is compared with other methods and evaluated using four representative iris image databases. Experiments show that our proposed method has excellent performance on iris localization.

Although the proposed method has many advantages above, it also has some defects. The serial network design of first localization and then segmentation increases the time for the model to process those two tasks; moreover, the method of optimizing radius regression with *L*_1_ loss does not take into account the pairing of the inner and outer radius of the iris. Finally, although the model has achieved optimization on most indicators, the segmentation effect on some data sets still needs to be improved.

In the future, we will explore more efficient approaches to improving segmentation performance on irregular iris image captured under visible illumination, and consider optimizing the radius regression loss function; moreover, we will consider optimizing the network structure to deal with iris localization and iris segmentation tasks in parallel to reduce the total process time.

## Figures and Tables

**Figure 1 sensors-23-02238-f001:**
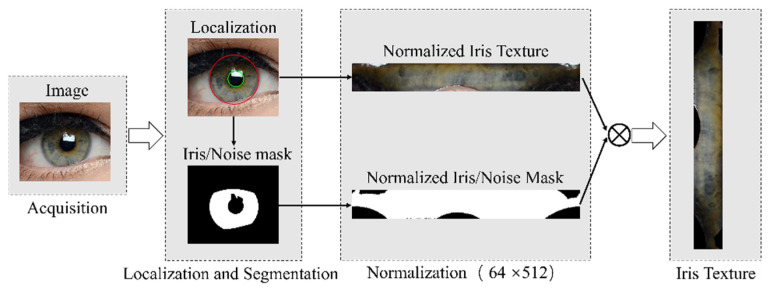
Iris-image preprocessing for the iris-recognition system, ⊗ presents the pixel-wise multiplication.

**Figure 2 sensors-23-02238-f002:**
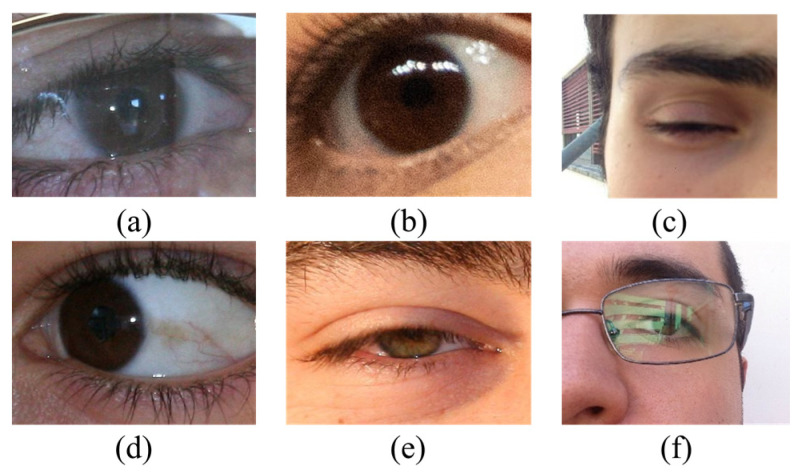
Example of degraded iris images: (**a**) glasses occlusion, (**b**) blur, (**c**) absence, (**d**) gaze deviation, (**e**) eyelid occlusion, and (**f**) specular reflection.

**Figure 3 sensors-23-02238-f003:**
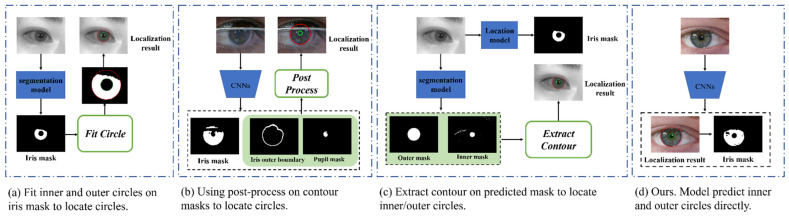
Illustrations of different methods for iris localization.

**Figure 4 sensors-23-02238-f004:**
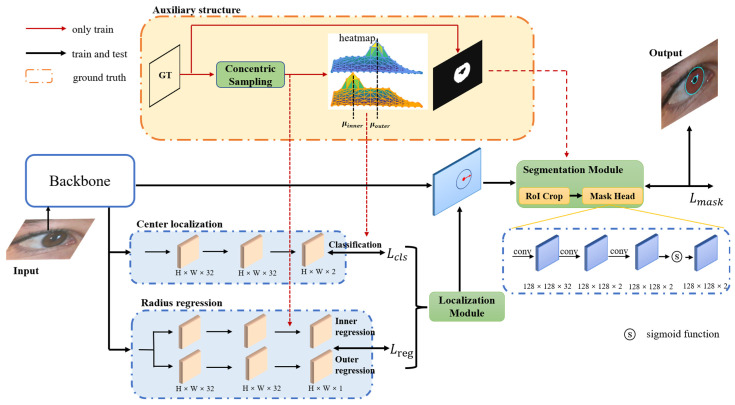
Architecture of ICSNet. Subnets in the three blue boxes are the center localization head, radius regression head, and mask head. Subnets in the orange box are the concentric sample operations, red arrow is the data flow, and our concentric sampling strategy is completely cost-free in inference time because auxiliary structures only exist during training. The localization module includes the classification and regression heads, which predict the circle bounding box. The mask module includes crop operation and mask head, which are responsible for cropping the iris region from the feature map and producing the segment iris mask.

**Figure 5 sensors-23-02238-f005:**
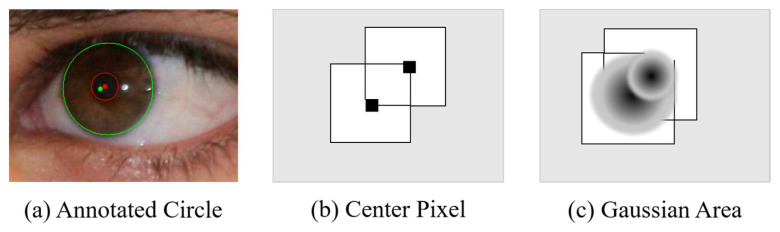
(**a**) Example of an iris, (**b**) inner-/outer-circle center, (**c**) and its Gaussian distribution. The centers of the inner and outer circles are highlighted in red and green, respectively. The Gaussian distributions of the inner and outer circles are shown in (**c**); the darker the color, the greater the value.

**Figure 6 sensors-23-02238-f006:**
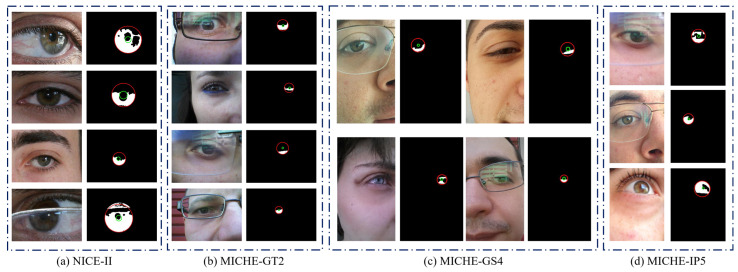
Example images and corresponding ground truths (including iris inner boundary (green), iris outer boundary (red), and iris mask (white)) of four iris databases.

**Figure 7 sensors-23-02238-f007:**
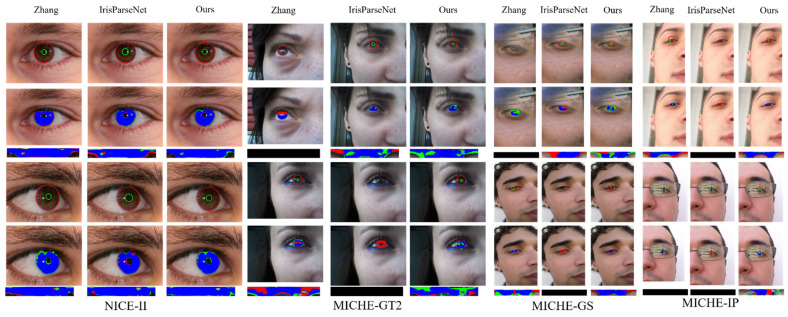
The results of the different methods on NICE-II [31] and MICHE [32]. For non-normalized and normalized masks, blue indicates true positive pixels, green represents false positive pixels, red represents false negative pixels. If locating the inner and outer circles fails, then the normalized iris result will be empty, which is indicated by a black bar image.

**Table 1 sensors-23-02238-t001:** Ablation study of concentric sampling. “↑” denotes the improvement of localization performance after using concentric sampling strategy to auxiliary training.

Data Sets	None	DC-CGR	$mIoU_{inner}^{box}$	$mIoU_{outer}^{box}$	$↑mIoU_{avg}^{box}$
NICEII	√		73.99%	94.06%	−4.11%
		√	69.40%	90.43%	
MICHE-GS	√		59.01%	80.66%	5.29%
		√	64.06%	86.20%	
MICHE-IP	√		58.01%	82.65%	4.87%
		√	63.07%	87.34%	
MICHE-GT2	√		54.30%	79.65%	4.79%
		√	57.47%	86.06%	

**Table 2 sensors-23-02238-t002:** Accuracy of iris segmentation and localization on NICE-II.

Methods	$mIoU_{in}^{box}$	$mIoU_{out}^{box}$	$mIoU_{avg}^{box}$	$mHdist^{in}$	$mHdist^{out}$	$E1^{mask}$	$mIOU^{mask}$	$E1^{norm}$
FCEDN	0.5564	0.9083	0.7323	0.0298	0.0175	0.0108	0.7246	0.2323
IrisDenseNet	0.5761	0.9107	0.7434	0.0279	0.0174	0.0103	0.7320	0.2243
FRED-Net	0.6373	0.9034	0.7703	0.0238	0.0184	0.0122	0.7276	0.2112
IrisParseNet	0.7026	0.9019	0.8023	0.0151	0.0185	0.0096	0.8751	0.1284
NIR-Zhang	0.7308	0.9394	0.8351	0.0149	0.0134	0.0072	0.9013	0.0909
ICSNet (ours)	0.7399	0.9406	0.8402	0.0135	0.0102	0.0079	0.8915	0.0913

**Table 3 sensors-23-02238-t003:** Accuracy of iris segmentation and localization on MICHE.

Methods	Data Sets	$mIoU_{in}^{box}$	$mIoU_{out}^{box}$	$mHdist^{in}$	$mHdist^{out}$	$E1^{mask}$	$mIOU^{mask}$	$E1^{norm}$
FCEDN	GS	0.2358	0.7743	0.0469	0.0268	0.0062	0.6067	0.3457
	IP	0.1735	0.7651	0.0518	0.0276	0.0046	0.6186	0.3152
	GT2	0.0849	0.6488	0.1053	0.0687	0.0100	0.5240	0.3827
IrisDenseNet	GS	0.3858	0.7733	0.0328	0.0346	0.0055	0.6956	0.3018
	IP	0.2317	0.7668	0.0409	0.0409	0.0070	0.6498	0.3013
	GT2	0.3998	0.7595	0.0434	0.0415	0.0043	0.7077	0.2882
FRED-Net	GS	0.5628	0.7710	0.0306	0.0314	0.0054	0.6388	0.2717
	IP	0.4014	0.7799	0.0360	0.0325	0.0052	0.6404	0.2440
	GT2	0.3767	0.7532	0.0474	0.0442	0.0048	0.7148	0.2418
IrisParseNet	GS	0.5353	0.7640	0.0650	0.0743	0.0046	0.7362	0.2545
	IP	0.5290	0.7071	0.0740	0.0875	0.0052	0.6823	0.2753
	GT2	0.5321	0.7771	0.0439	0.0519	0.0041	0.7504	0.1867
NIR-Zhang	GS	0.5327	0.8058	0.0408	0.0326	0.0051	0.6919	0.2605
	IP	0.5369	0.8436	0.0245	0.0224	0.0037	0.7429	0.2178
	GT2	0.5930	0.8516	0.0265	0.0260	0.0053	0.7039	0.2118
ICSNet (ours)	GS	0.6406	0.8620	0.0132	0.0162	0.0050	0.7001	0.2465
	IP	0.6307	0.8734	0.0165	0.0186	0.0036	0.7431	0.2099
	GT2	0.5747	0.8606	0.0381	0.0227	0.0061	0.6857	0.2319

**Table 4 sensors-23-02238-t004:** Comparison of parameters and GFLOPs.

Methods	Parameters	GFLOPs
FCEDN	115.18 M	62.65
IrisDenseNet	142.95 M	75.67
IrisParseNet	125.18 M	87.03
NIR-Zhang	95.61 M	50.12
ICSNet (ours)	69.53 M	47.01

## Data Availability

Not applicable.

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
