# Peer review of "Double-Center-Based Iris Localization and Segmentation in Cooperative Environment with Visible Illumination"

_sensors, 2023, doi:10.3390/s23042238_

Round 1
Reviewer 1 Report
The manuscript proposes a method for iris localization based on double-center based end-to-end iris localization and segmentation network. To facilitate efficient training, a concentric sampling strategy is introduced according to the center distribution of the inner and outer iris circles. I suggest authors bring the following points into their consideration to revise the manuscript accordingly
- The abstract should be revised as it does not enough chiefly introduce the area of research along with the research question.
- The introduction should be revised in more professional way. The contribution of this manuscript should be clearly listed. Also, outline the rest of paper at the end of introduction.
- Related work part (2. Related Research) is weak, which make it is difficult to identify novel points in Iris localization. Thus, the related work should be enriched by discussing more published articles, Recommended works:
- https://doi.org/10.1007/s12559-022-10065-9
- There is no difference between the Iris localization or iris segmentation, thus no need to divide it to two sections.
- Please, explain the proposed method in more details. What is the novelty of the proposed method compared to the state of the art?. It is not clear how eye field(region)is detected or localized.
- All images shown in the description of the proposed method are for clear eye image without noise.
- No details about size of input image or preprocessing
- In the experiments, please give descriptions for datasets with sample images showing challenges the proposed method focuses on.
- Conclusion should highlight the achievements. Here it is more or less similar to the Abstract. Make sure conclusions reflect on the strengths and weaknesses of the work, how others in the field can benefit from it and thoroughly discus future work.
Reviewer 2 Report
My comments are marked on the manuscript I have uploaded. On page 6, the top line was partially cropped by my scanner. It said "The notation in equation (4) does not make sense, because the ...".

Author Response
Please see the attachment.
Thanks to the reviewer's careful review of the paper, we have corrected some of the language and illustration errors you pointed out. For other information that needs to be replied, please refer to the attached PDF.

Round 2
Reviewer 1 Report
All my concerns have been considered. Thus, I have no further comments.